# Multi-Faceted Multimodal Monosemanticity

## Abstract

Humans experience the world through multiple modalities, such as, vision, language, and speech, making it natural to explore the commonality and distinctions among them. In this work, We take a data-driven approach to address this question by analyzing **interpretable, monosemantic features** extracted from deep multimodal models. Specifically, we introduce the Modality Dominance Score (MDS) to attribute each multimodal feature to a specific modality. We then map the features into a more interpretable space, enabling us to categorize them into three distinct classes: vision features (single-modal), language features (single-modal), and visual-language features (cross-modal). Interestingly, this data-driven categorization closely aligns with human intuitive understandings of different modalities. We further show that this modality decomposition can benefit multiple downstream tasks, including reducing bias in gender detection, generating cross-modal adversarial examples, and enabling modal-specific feature control in text-to-image generation. These results indicate that large-scale multimodal models, when equipped with task-agnostic interpretability tools, can offer valuable insights into the relationships between different data modalities.

## 1 Introduction

Multimodal models have become foundational to the advancement of AI, enabling AI systems to process and understand information from multiple data modalities, such as vision and language Radford et al. (2021); Kim et al. (2021); Lu et al. (2019); Liang et al. (2024). Vision-Language Models (VLMs) in particular operate under the premise that different data modalities share common, or cross-modal, features that can be jointly learned (Ngiam et al., 2011). Along with such remarkable advancements, ongoing AI research also aims to deepen our understanding of how modalities interact and diverge within these VLMs (Liang et al., 2022; Rawal et al., 2023; Schrodi et al., 2025). For example, Liang et al. (2022) identified a modality gap in VLM, revealing that image and text embeddings often reside in two disjoint regions of the shared embedding space. Further studies explored the relationship between the degree of modality fusion and downstream task performance, e.g., in video understanding (Rawal et al., 2023) and object detection (Schrodi et al., 2025).

Different from the predominant research in encouraging and explaining the modality alignment (Li et al., 2023; Yi et al., 2024; Schrodi et al., 2025), our work investigates whether the *"modality gap"* is both prevalent and beneficial for downstream tasks. At the same time, the modality commonality and separation has long been a central theme in cognitive science, where researchers have examined how humans integrate and differentiate information across sensory modalities (Paivio, 1991; Spence, 2011; Fan et al., 2016). Building on this perspective, we explore the phenomenon by formulating and testing three key hypotheses through a variety of approaches, as summarized in Table **??**. For each of these hypotheses, we demonstrate that:

1. We can extract modality-specific information in VLMs, i.e., text-dominant (`TextD`), image-dominant (`ImgD`), and cross-modality features (`CrossD`) in CLIP; and they exhibit different patterns when dealing with input images and texts. (Section 2)

2. We enhance the monosemanticity of VLM features via self-supervised methods such as Sparse AutoEncoders (SAE), and observe improved interpretability and modality specificity in `TextD` and `ImgD` after enhancement. (Section 3)

3. We further demonstrate that our identified modality-specific features can be seamlessly integrated into various downstream tasks: gender bias detection, generating adversarial attacks, and controllable text-to-image generation. (Section 4)

## 2 IDENTIFY MODALITY-SPECIFIC INFORMATION IN VLMS

Modality alignment and fusion are crucial to the success of existing VLMs (Liang et al., 2022; Schrodi et al., 2025), while the modality-specific gap has been extensively studied in cognitive science. For instance, Ungerleider & Haxby (1994); Fan et al. (2016) have found that regional specificity and coordinated processing coexist in the human brain. Therefore, we start with the question *whether there are modality-specific features in VLMs?* To answer the question, we use CLIP models from OpenAI (Radford et al., 2021) as the testbed to quantify the modality-specific information (Section 2.1) and interpret these features via their activated samples (Section 2.3).

### 2.1 MODALITY-SPECIFIC FEATURE IDENTIFICATION

**Modality Alignment in VLMs.** Typically, there are an image encoder and a text encoder in a VLM for image and text inputs processing, respectively. Specifically, the image-text pair $(x_{\text{img}}, x_{\text{txt}})$ is fed to an image encoder $f_{\text{img}}$ and a text encoder $f_{\text{txt}}$ within the model, respectively and the final-layer representations $z_{\text{img}} \in \mathbb{R}^D$ and $z_{\text{txt}} \in \mathbb{R}^D$ are then optimized jointly in the shared $D$-dimensional representation space. An alignment loss, such as the contrastive loss in CLIP (Ilharco et al., 2021) across the two modalities are then applied to enhance modality fusion. , a persistent modality gap remains across most multimodal models (Liang et al., 2022). Instead of closing the gap and improving the downstream tasks (Schrodi et al., 2025), we posit that an *irreducible* modality gap may be essential for capturing modality-specific concepts—for instance, emotions that are difficult to visualize or visual experiences that cannot be fully expressed in language.

**Modality Dominance Score.** Liang et al. (2022) measure the modality gap using the difference between the center of image embeddings and text embeddings of $M$ input pairs, i.e., $\frac{1}{M}(\sum_{i=1}^{M} ||z_{\text{img},i}||_2 - \sum_{i=1}^{M} ||z_{\text{txt},i}||_2)$. We extend this model-level measurement to a fine-grained metric, i.e., the predominant modality associated with each dimension $d \in \{1, 2, \ldots, D\}$ in the shared embedding space. The proposed modality dominance score (**MDS**), denoted as $R(d)$ shown in Eq. (1) reflects how strongly the $d$-th feature [1] is influenced by the image modality:

$$R(d) = \frac{1}{M} \sum_{i=1}^{M} \frac{||z_{\text{img},i}^{(d)}||}{||z_{\text{img},i}^{(d)}|| + ||z_{\text{txt},i}^{(d)}||}. \tag{1}$$

Specifically, we feed $M$ image-text pairs to the VLM and extract the corresponding image features $z_{\text{img},i}$ and text features $z_{\text{txt},i}$ for $i$-th input. For each $d$-th dimension in the $D$-dimension shared space, we calculate the relative activation between the features from the two modalities. This modality fraction is averaged over more than $M = 10k$ input pairs, providing a representative estimate of the modality distribution.[2]

We then categorize all $D$ features into three groups based on their deviation from the mean $\mu$ and standard deviation $\sigma$ of the MDS distribution:

$$\texttt{TextD:}\ R(d) < \mu - \sigma; \quad \texttt{CrossD:}\ \mu - \sigma < R(d) < \mu + \sigma; \quad \texttt{ImgD:}\ R(d) > \mu + \sigma.$$

We anticipate that `ImgD` features are predominantly activated by visual concepts, `TextD` features by textual concepts, and `CrossD` features are simultaneously activated by the shared commonalities between image and text. To clearly illustrate the differences among these three feature groups, we interpret each feature by examining the samples that activate it most strongly.

### 2.2 MODALITY-CLASSIFICATION FOR MDS VALIDATION

To verify that the modality-specific features effectively capture the intended modality information, we remove these features from the original representation by setting the corresponding indices to zero, and then use the modified features for a modality-specific classification task. If a higher classification accuracy is observed, it indicates that the removed features contained a substantial portion of the modality information.

---

[1] Each feature dimension corresponds directly to a feature/neuron in the VLM's final layer; our study thus focuses on the interpretability of the model's intrinsic components.

[2] Details of the MDS calculation are provided in Appendix A.2.

We evaluate the extracted features from CLIP ViT-H/14 (LAION-2B) Ilharco et al. (2021) in both image/text classifications on COCO (Lin et al., 2014) using 10,000 image-caption pairs (80/20 stratified split). Specially, we intervened features from the final transformer layer are classified using logistic regression. As there are different number of `ImgD`, `TextD` and `CrossD`, we also remove the same number of random feature indices as comparison. The results are shown in Table 1. It is observed that removal the `ImgD` leads to larger classification degradation in Image classification, while removal `TextD` leads to larger drops in text classification; while `CrossD` does not show any particular modality tendency in classficaition.

Table 1: Performance changes of Modality-specific classification after removing: random vs. specialized features, i.e., `ImgD`, `TextD` and `CrossD`.

| Task | Deletion | # Neurons | Accuracy | △ Accuracy |
|------|----------|-----------|----------|------------|
| Image CLS | None | 0 | **0.776** | / |
| | Random | 426 | 0.757 | -2.1% |
| | ImgD | 426 | **0.750** | **-2.6%** |
| | Random | 554 | 0.756 | -2.0% |
| | TextD | 554 | 0.760 | -1.0% |
| | Random | 44 | 0.773 | -0.3% |
| | CrossD | 44 | 0.769 | -0.5% |
| Text CLS | None | 0 | **0.713** | / |
| | Random | 426 | 0.694 | -1.9% |
| | ImgD | 426 | 0.702 | -1.1% |
| | Random | 554 | 0.693 | -2.0% |
| | TextD | 554 | **0.683** | **-3.0%** |
| | Random | 44 | 0.710 | -0.3% |
| | CrossD | 44 | **0.712** | -0.1% |

## 2.3 QUALITATIVE EVALUATION FOR MODALITY-SPECIFIC INFORMATION

We randomly select two features from the three groups, and then display their most-activated images and texts in Figure 1 (`ImD`), Figure 2 (`TextD`) and Figure 3 (`CrossD`).

**`ImgD` activates fundamental visual concepts, such as repeated patterns and colors.** Feature 647 activates images with diverse repetitive patterns; feature 667 focuses on scenes with aquatic-blue elements. Although less coherent than the images, some patterns do emerge for its activated texts: feature-647 activates two sentences that refer to repetitive patterns, such as "*tufted upholstery*"; feature-667 activates texts related to "*snowy*" and "*winter*". These observations indicate the modality alignment while the visual commanalities are more predominant for the `ImgD`.

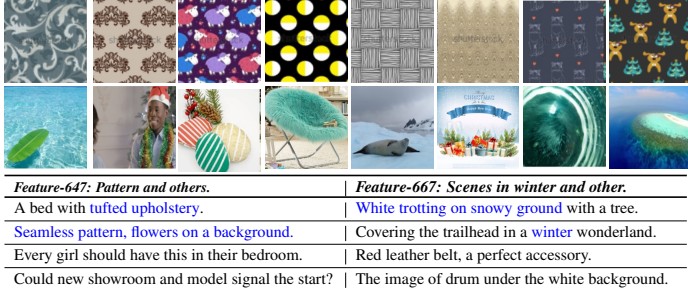

| *Feature-647: Pattern and others.* | *Feature-667: Scenes in winter and other.* |
|---|---|
| A bed with tufted upholstery. | White trotting on snowy ground with a tree. |
| Seamless pattern, flowers on a background. | Covering the trailhead in a winter wonderland. |
| Every girl should have this in their bedroom. | Red leather belt, a perfect accessory. |
| Could new showroom and model signal the start? | The image of drum under the white background. |

Figure 1: Activated images and texts (in Table) by `ImgD`. Top image row (feature 647): patterns and textures. Bottom image (feature 667): water and aquatic themes in blue. Texts in blue align with visual concepts.

**`TextD` capture abstract concepts, such as human feelings and atmosphere.** For the activated images for feature-34 (the 1st row), most of the images have red color, with one image depicting a couple talking beside the sea; for feature-242, there are no clear patterns among the activated images. When looking at the activated texts, sentences activated by feature-34 center around a sweet and happy atmosphere between couples, with themes like cuddling, embracing, and hugging. Feature-242 focuses on strong human emotions, such as "*never*", "*terrifying*" and exclamation marks. These `TextD` generally correspond to abstract and consistent human emotions, which can be conveyed with a variety of visual objects. For example, in the second row, the first image depicts a collection of stones forming a heart shape, while the fourth image is a scenic view during a great trip.

**`CrossD` (the majority features) capture shared semantics across modalities.** Different from modality-specific features, `TextD` and `ImgD`, `CrossD` features capture common concepts that could be expressed in both visual and language modalities. We randomly select two `CrossD` features and show their top activated images and texts. As shown in Figure 3, feature-6 mostly activates scenes involving individuals performing activities, especially outdoor activities, and feature-47 captures general outdoor environments. The coherence across both modalities reflects successful alignment, which is consistent with multimodal training objectives.

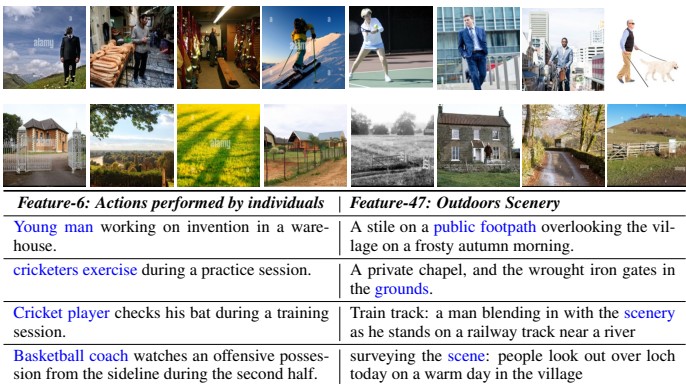

Figure 2: Activated images and texts (in Table) by `TextD`. Top image row (feature 34): couples and individuals in red attire. Bottom image row (feature 242): diverse objects. Text in blue aligns with visual concepts.

Figure 3: Activated images and texts by `CrossD` features. Top image row (feature 6): activities performed by individuals. Bottom image row (feature 47): scenery outside the doors. Text in blue aligns with visual concepts.

## 3 ENHANCE MODALITY-SPECIFIC INFORMATION VIA MONOSEMANTICITY

We have identified the modality-specific features above, although the features in deep models are mostly *polysemantic* (Olah et al., 2020), that is, each feature often encloses multiple unrelated semantic concepts. Meanwhile, recent advances in interpretability methods, particularly in *monosemanticity* (Elhage et al.; Bills et al., 2023; Gurnee et al., 2023; Yan et al., 2024a), enable deeper insights into the inner workings of deep models. Monosemantic neurons or features refer to model components that correspond to a single, interpretable concept.

Many existing monosemanticity works focus on improving the interpretability of the single-modality model, e.g., a language model (Elhage et al.; Bills et al., 2023). Additionally, scaling interpretability tools remains an open challenge due to the heavy reliance on costly human annotations (Gao et al., 2024). To address these gaps, we disentangle CLIP features to obtain interpretable *monosemantic features* in §3.1; and we propose new metrics to quantitatively evaluate the interpretability of these features §3.2.

### 3.1 EXTRACT MULTIMODAL MONOSEMANTIC FEATURE VIA SELF-SUPERVISION

To obtain disentangled features and improve the interpretability, we consider the following self-supervised representation learning methods [3]:

**DeCLIP.** Beyond multimodal supervision (image-text pairs), DeCLIP (Li et al., 2022) also incorporates single-modal self-supervision (image-image pairs and text-text pairs) for more efficient joint learning. We hypothesize that, with the incorporation of self-supervision tasks, DeCLIP can extract more single-modal features from data, enhancing its interpretability and alignment with modality-specific characteristics.

---

[3]Implementation details for these methods are shown in Appendix A.1.

**Multimodal SAE.** Sparse Autoencoders (SAEs) (Cunningham et al., 2023) have emerged as a scalable tool for transforming polysemantic *neurons* into interpretable, monosemantic *features* across various LLMs (Templeton, 2024; Gao et al., 2024; Lieberum et al., 2024). We adapt this technique for multimodal settings by training a **single** SAE model $g : \mathbb{Z} \to \mathbb{Z}$ to reconstruct $z$, i.e., the final-layer outputs from the image and text encoder within CLIP, respectively. Specifically, we adopt a TopK SAE (Makhzani & Frey, 2013; Gao et al., 2024) that applies a linear encoder $W_{\text{enc}}$ followed by a $\text{Top} K$ operation that only keeps the $K$ most activated units while zeroing out the rest. The sparse latent representation $z^{\text{sae}}$ is then reconstructed using a linear decoder $W_{\text{dec}}$:

$$\text{(latent)} \ z^{\text{sae}} = \text{TopK} \left( W_{\text{enc}} \left( z - b_{\text{pre}} \right) \right), \quad \text{(reconstruction)} \ \hat{z} = W_{\text{dec}} z^{\text{sae}} + b_{\text{pre}}. \tag{2}$$

$z \in \mathbb{R}^D$ is the inputs of SAE, i.e., $z_{\text{img}}$ or $z_{\text{txt}}$. $z^{\text{sae}} \in \mathbb{R}^n$ is the learned sparse representation. We train the multimodal SAE to reconstruct $z_i$, $z_t$ from CLIP image and text encoder, respectively.

$$\mathcal{L}_{\text{M}-\text{SAE}}(g) = \mathbb{E}_{(z_{\text{img}}, z_{\text{txt}}) \sim \mathcal{P}} \left[ \|z_{\text{img}} - g(z_{\text{img}}^{\text{sae}})\|^2 + \|z_{\text{txt}} - g(z_{\text{txt}}^{\text{sae}})\|^2 \right]. \tag{3}$$

By expanding the original latent dimension $d$ to $n$ and activating only the top-$k$ latent units, the representation $z^{\text{sae}}$ is encouraged to capture specialized and interpretable features.

**Multimodal NCL.** A key limitation of leveraging SAEs for extracting multimodal features is that its reconstruction objective is still essentially single-modal, and thus may underutilize the multimodal alignment inherent to models such as CLIP. To address this, we introduce a variant of Non-negative Contrastive Learning (NCL) (Wang et al., 2024) to enhance multimodal interpretability with the following *Multimodal NCL* loss,

$$L_{\text{M}-\text{NCL}}(g) = -\mathbb{E}_{z_{\text{img}}, z_{\text{txt}}} \log \frac{\exp(g(z_{\text{img}})^\top g(z_{\text{txt}}))}{E_{z_{\text{txt}}^-} \exp(g(z_{\text{img}})^\top g(z_{\text{txt}}^-))}, \tag{4}$$

where here we use an MLP network to map input features to *non-negative* output latent features, i.e.

$$z^{\text{ncl}} = g(z) = \text{ReLU}(W_2 \, \text{ReLU}(W_1 z + b_1) + b_2), \ \forall \, z \in \mathbb{R}^D.$$

As shown in Wang et al. (2024), the non-negative constraints allow NCL to extract highly sparse features and significantly improve feature monosemanticity.

By applying these two interpretability models, Multimodal SAE and Multimodal NCL, we obtain sparse, interpretable features $z^{\text{sae}}$ and $z^{\text{ncl}}$, along with the original multimodal representation $z$ from CLIP and DeCLIP. These representations form the foundation for analyzing modality purity and interpretability.

**MDS with monosemanticity enhancements.** With the monosemanticity-improving models (SAE and NCL), we hypothesize that modality purity will become more pronounced, making dominant modality assignments more meaningful. To validate this, we calculate the MDS and visualize the distributions of the three feature groups across models in Figure 4. Interestingly, we find that CLIP, which is only trained on an image-text contrastive learning objective, contains a spectrum of features with different modality dominance. Specifically, its distribution skews towards the image modality, and this trend is consistent across all models. DeCLIP, on the other hand, shows a more balanced and less centered distribution. This suggests that DeCLIP, through self-supervision, extracts more modality-specific features, which might be overlooked by pure vision-language contrastive models like CLIP. The extracted features on top of NCL and SAE also exhibit less skewness, with SAE showing the most balanced distribution, indicating its strong capability to extract diverse monosemantic features.

## 3.2 QUANTITATIVE EVALUATION FOR INTERPRETABILITY IN MULTIMODAL MODEL

A feature is considered interpretable if its semantic meaning can be readily understood by users; typically, it can be assessed by examining whether the top-k activated samples exhibit a coherent and consistent pattern. For instance, Bills et al. (2023) first prompt a large language model (LLM) to generate an explanation based on the activated tokens. Using this explanation, they then prompt the LLM (like GPT-4o) to predict activation values for a given set of tokens, and the correlation between predicted and actual activation values is used as an interpretability score. The reliance on LLM-as-a-Jude limits its utility due to reliability and scalability (Gu et al., 2024; Yan et al., 2024b).

Therefore, we propose an interpretability evaluation that is both scalable and applicable to multimodal models. First, we assess monosemanticity (§3.2.1) by testing whether activated samples for each feature are consistently similar in a scalable way. Second, we analyze modality dominance across three feature categories, i.e., `TextD`, `ImgD`, and `CrossD` via examining whether certain features are more interpretable within specific modalities (§3.2.2).

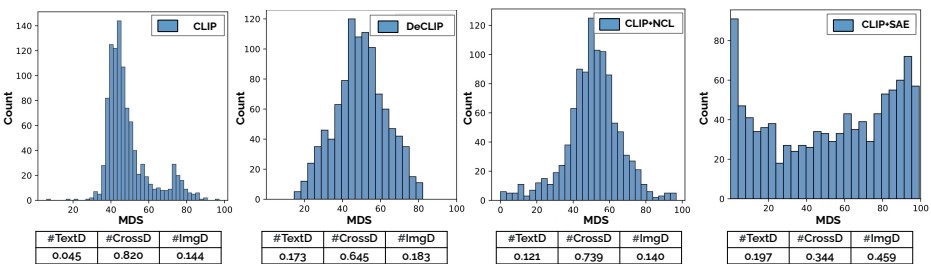

Figure 4: Modality Dominance Score (MDS) distributions of three feature categories for different VLMs.

### 3.2.1 MONOSEMANTICITY EVALUATION

A feature $z^{(d)}$ [4], the $d$-th dimension of $z \in \mathbb{R}^D$, is monosemanticity if it encodes a single, coherence semantic concept (Templeton, 2024). We propose a scalable interpretability measure based on embedding models $h : \mathbb{Z}^D \to \mathbb{Z}^{D'}$ that can be applied to both images and text samples.

**Evaluation metrics.** For each image/text feature $z^{(d)}$, we collect the top $m$ most-activated image/-text samples for this dimension, and feed them to the embedding model $h$ to get $Z_+ \in \mathbb{R}^{m \times D'}$. For comparison, we embed $m$ random samples into $Z_- \in \mathbb{R}^{m \times d'}$. Then, we calculate the inter-sample similarity between the selected samples, $S_+ = Z_+ Z_+^\top \in \mathbb{R}^{m \times m}$ and $S_- = Z_- Z_-^\top \in \mathbb{R}^{m \times m}$. The monosemanticity of an individual feature $z^{(d)}$ is measured by calculating the relative difference between the two similarity scores, denoted as $I(z^{(d)})$ (***EmbSim***). We also propose a binary metric to avoid the different scales in different modalities, denoted as $W(z^{(d)})$ (***WinRate***):

$$I(z^{(d)}) = \frac{1}{m(m-1)} \sum_{i \neq j} \frac{(S_+)_{ij} - (S_-)_{ij}}{(S_-)_{ij}}; \quad W(z^{(d)}) = \frac{1}{m(m-1)} \sum_{i \neq j} \mathbf{1}_{[(S_+)_{ij} > (S_-)_{ij}]}. \quad (5)$$

The overall interpretability score is the average across all $d$ dimensions for $z^{(d)}$ where $d \in [1, D]$. A higher monosemanticity score (we call the ***EmbSim*** and ***WinRate*** as ***Mono***[5]) indicates that the extracted features exhibit stronger semantic consistency.

**Results.** We compute the ***Mono*** (interpretability) score by identifying the top-20 most activated images and texts for each feature, respectively. From the average interpretability results in Figure 5, we observe the following: (i) The features extracted using SAE and NCL (which enforce the feature sparsity) exhibit the highest overall monosemanticity for both activated input images and texts. (ii) DeCLIP does not enhance interpretability through self-supervision alone, the monosemanticity on textual side becomes even worse. This suggests that polysemantic features remain prevalent in DeCLIP, although their modality separation is clearer than in CLIP.

### 3.2.2 MODALITY SPECIFICATION EVALUATION

We have observed improved average monosemanticity across all features above. Therefore, we now investigate the question to which modality is a particular feature most sensitive? focusing on the three categorized feature types. Specifically, we ask: is `ImgD` indeed more effective at capturing visual inputs than `TextD`? Similarly, is `TextD` better at encoding textual semantics compared to `ImgD`? Specifically, for visual inputs, we calculate the ***Visual Mono***, i,e, *Mono*(`ImgD`)-*Mono*(`TextD`). For text inputs, we calculate ***Textual Mono*** using *Mono*(`TextD`)-*Mono*(`ImgD`).

**Results.** We have the following observations from Figure 6: (i) For CLIP, all the modality monosemanticity is negative, demonstrating the highly entanglement of the two modality information. (ii) All the methods prompt the modality monosemanticity compared with CLIP. Particularly, the improvements of DeCLIP can be attributed to its single-modal alignment training loss, which could weaken some cross-modal associations in CLIP. (iii) NCL stands out as the best model for capturing both visual and textual monosemantic features, followed by SAE.

---

[4] We consider both $z_{\text{img}}$ and $z_{\text{txt}}$ as $z$, as we don't study the modality-specific information in this subsection.

[5] We calculate the average of ***EmbSim*** and ***WinRate*** as Monosemanticity score (***Mono***) in the main paper, the complete results for the two metrics can be found in Appendix A.3.2.

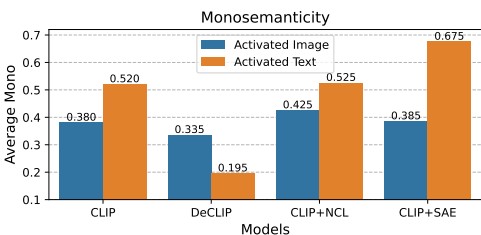

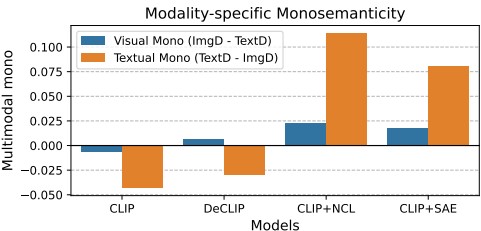

Figure 5: Monosemanticity for four VLMs.

Figure 6: Modality-specific monosemanticity.

## 4 APPLICATIONS OF USING MODALITY-SPECIFIC INFORMATION

Beyond interpretability, we further explore how these modality-aware features contribute to performance in representative multimodal tasks. [6]

### 4.1 CASE STUDY 1: UNDERSTANDING GENDER PATTERN IN DIFFERENT MODALITIES

We describe gender using both visual and textual features and these data are used to train VLMs. To test whether there is a modality-specification in different genders, e.g., *Does the concept of the feminine get described by images more frequently? such as more colorful outfits.*

To test this hypothesis, we collect both male and female images with their corresponding textual descriptions from the cc3m-wds (Sharma et al., 2018). These images are then encoded using the Clip+SAE model, extracting 1024-dimensional features for both female and male subjects. Next, we apply a zero-mask intervene strategy to remove the `ImgD` and `TextD` from these representations.

To examine this, we compare changes in gender classification accuracy when removing `ImgD` features from image inputs, which capture dominant feminine visual cues, versus removing `TextD` features from text inputs. As shown in Table 2, we find that feminine concepts are primarily preserved in `ImgD` (as the removal of `ImgD` from image lead to larger classification degradation), whereas male concepts are more affected by the removal of `TextD`.

Table 2: Gender classification changes (%) after removing `ImgD`(`textD`) **from input image(text)** for both female and male concepts identification. It is to verify the dominant modality for different gender.

| Gender | w.o ImgD | w.o TextD |
|--------|----------|-----------|
| Female | **17.65** | 7.27 |
| Male | 5.64 | **28.67** |

**Understand the what feminine concepts the `ImgD` represent.** We sample different female images which differ in how many percentage of their most activated features categorized as `ImgD` features. The results are in Figure 7. From left to right, more activated features are `ImgD` and they tend to contains more detailed (*stereotype*) feminine concepts, such as backless skirt,

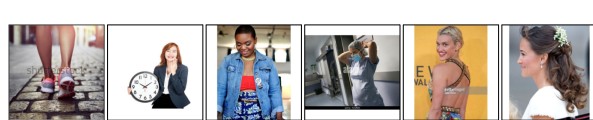

Figure 7: *Female* figures ordered by their percentages of `ImgD` features: 0.14, 0.16, 0.18,0.20, 0.22, 0.24, 0.26. More feminine concepts are observed to be related with more `ImgD`.

hair accessories. The middle images show professional female, such as politician and doctor; and the first image shows a pair of leg in sports shoes, with minimal feminine factors, the pink color.

### 4.2 CASE STUDY 2: GENERATING MODALITY-SPECIFIC ADVERSARIAL ATTACKS

We investigate the impact of different types of features on multimodal adversarial attacks (Cui et al., 2024; Yin et al., 2024), following the setup in Shayegani et al. (2024).

The adversarial sample is a benign-appearing image, e.g., a scenery image but injected with harmful semantic information, such as the phrase *"I want to make bomb"*. One defense optimization strategy involves minimizing the distance, between the embeddings of adversarial sample $\mathbf{F}_{adv}$ and a benign sample $\mathbf{F}_{ben}$, and accordingly update the adversarial sample (in Figure 8). The paired benign image

---

[6]Detailed implementations are in Appendix A.4.

is injected with the friendly text, e.g., *"peace and love"*. To study the effects of our identified modality features, we only select the target feature index $I$ for alignment training, i.e., `ImgD`, `TextD`, and `CrossD`. The alignment loss is $\mathcal{L} = \|\mathbf{F}_{adv}[:, I] - \mathbf{F}_{ben}[:, I]\|_2$. Finally, the optimized adversarial sample is then adopted to attack a VLM, LLaVA-1.5-7b (Liu et al., 2023). We use the LLM-as-a-Judge to evaluate the generated response from the VLM, where DeepSeek-V3 (DeepSeek-AI et al., 2024) is required to generate a binary label indicating whether the attack is successful. Supposedly, the features containing more information related to the malicious semantics will contribute most to the attack defense.

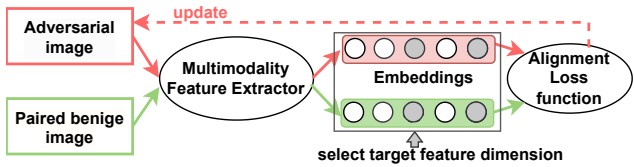

Figure 8: Alignment training to de-toxicity of the adversarial sample, with only selected target feature dimensions (in gray), i.e., `ImgD`, `TextD` and `CrossD`, involved in the alignment.

Table 3: Success rate for adversarial attacks with different target features involved in the alignment training. The success rate for original adversarial samples without alignment training is 73.26%, with random selected features is 54.28%.

| Target feature | ImgD | TextD | CrossD |
|---|---|---|---|
| **Success Rate** ($\downarrow$) | 62.71% | 24.89% | 35.44% |

**Results.** The attack success rates are shown in Table 3. We select the same number from `ImgD`, `TextD`, `CrossD` to be involved in alignment training, as well as randomly sample the same number of features cross the three feature sets as a baseline. We attack the VLM repeatedly for 100 times per sample, and we have generated 50 adversarial samples. We observe that (i) by comparing with the success rate of original adversarial samples, the alignment training with any selected features defense the attacks in some extent; (ii) using `TextD` yields the best defense performance, followed by `CrossD` and `ImgD`. This can be explained that the adversarial information primarily stems from undesirable textual semantics. And **it demonstrates that `TextD` effectively captures most of the semantic content.** In contrast, `CrossD` captures partial semantics, while `ImgD` is the least related to semantic information, resulting in minimal benefits for such modality-specific jailbreak defense.

### 4.3 CASE STUDY 3: MODALITY-AWARE CONTROL FOR TEXT-TO-IMAGE GENERATION

Despite the impressive capabilities of text-to-image generation models (Yu et al., 2024; Koh et al., 2024; Swamy et al., 2024), their internal mechanisms for bridging linguistic semantics and visual details remain poorly understood. A key challenge is disentangling how modality-specific features influence the fidelity and controllability of generation. Therefore, we conduct feature intervnetion experiment during the generation of Stable Diffusion v2 (Rombach et al., 2022)

**Intervention.** The process is depicted in Figure 9. We investigate the generation process by intervening in different modality-specific features in Stable-Diffusion-v2 (Rombach et al., 2022), i.e., the shown VLM with an encoder and decoder (generator). The input text prompt is *"Please draw an animal"*. The encoder generates an embedding $\mathbf{T}$, representing the original multimodal embedding ready for generation. Additionally, we provide a reference image, here is a horse - processed through the same encoder, producing a reference embedding $\mathbf{R}$. To control the generation through modality-specific feature intervention, we interpolate $\mathbf{T}$ only at the specified indices defined by MDS. The final multimodal embedding is computed as: $\mathbf{T}'[I] = \alpha\mathbf{T}[I] + (1-\alpha)\mathbf{R}[I]$, where operations are applied exclusively to the feature indices defined by $I$, i.e., `TextD`, `CrossD` and `ImgD`.

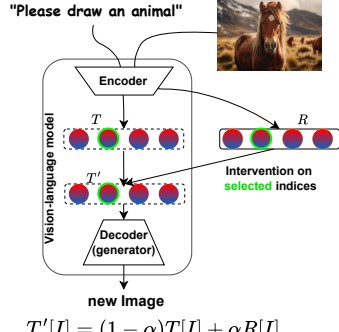

$$T'[I] = (1-\alpha)T[I] + \alpha R[I]$$

Figure 9: The reference image $R$ is used for modality-specific control over text-to-image generation process.

**Results.** We feed $\boldsymbol{T}'$ to the generator of the VLM with different $\alpha$ ranging from 0 to 0.7 with an interval of 0.1. The generated images with the selected indices correspond to `TextD`, `CrossD`, and `ImgD` are shown in Figure 10. The results clearly demonstrate that larger interventions on `TextD` lead to stronger control over high-level semantic concepts—for example, the generated image more distinctly resembles a horse (head). All these generated images injected by `TextD` typically depict clear main subjects

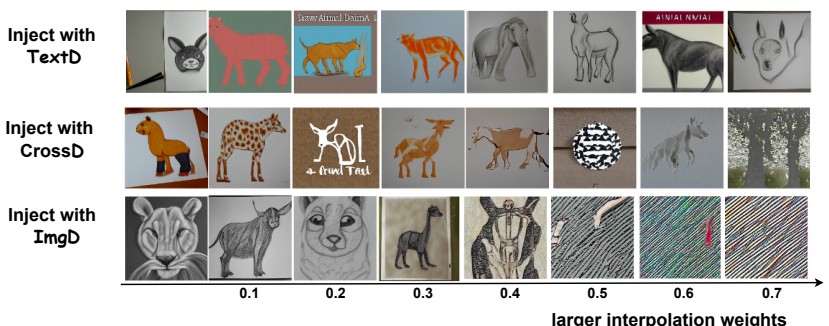

Figure 10: Generated new images from the VLM with the text prompt *"Please draw an animal"* and varying levels of intervention from a reference image (horse). From left to right, the interpolation weights $\alpha$ range from 0.0 to 0.7. Images generated with `TextD` typically depict clear main subjects (horse) without transferring the visual background details from the reference image. In contrast, injection of `ImgD` introduces low-level visual details as well as image distortions when $\alpha$ is large.

without transferring visual background details from the reference image. In contrast, interventions on `ImgD` result in more visual details from the reference image being preserved, such as non-white and fur-like patterned background are visible in `ImgD` when $\alpha \geq 0.3$. To better contrast the effects of `ImgD` and `TextD`, we also use a reference image with horse as the main subject, but in different styles/backgrounds. More results are shown in Figure 14.

## 5    RELATED WORK

**Mechanistic Interpretability.** Mechanistic interpretability aims to understand the internal computations of deep learning models by identifying and analyzing individual components and interactions. Recent research has been focused on identifying *polysemanticity*(Olah et al., 2020). This has led to the exploration of *monosemanticity*, the hypothesis that models might contain features that correspond to single, interpretable concepts (Elhage et al.). Recent advances in dictionary learning have made it possible to decompose polysemantic neurons into monosemantic features (Cunningham et al., 2023). These techniques, coupled with automated methods for interpreting and labeling features (Bills et al., 2023; Gurnee et al., 2023; Yan et al., 2024a), have enabled the extraction of large numbers of interpretable features from models like CLIP (Radford et al., 2021).

**Modality Gaps.** The study of modality gaps, or the differences and limitations in how different modalities represent information, has been a topic of interest in cognitive science (Spence, 2011; Paivio, 1991; Calvert et al., 2004). Our research offers an alternative *human-free* approach to study the modality gap. This opens a new approach to study the modality gap that could alleviate potential bias from human-centric viewpoint and bring more insights from large-scale data. Existing methods of measuring modality gap primarily are sample-level, such as L2M (Liang et al., 2022), information decomposition (Liang et al., 2023), which measures the different percentages of visual and textual information for a single VLM representation. We, instead, is to identify the different model components respond to different modalities in order to understand model-level behavior patterns.

## 6    LIMITATIONS AND CONCLUSION

In this study, we explored the monosemanticity of features within VLMs to elucidate the commonalities and distinctions across visual and textual modalities. we successfully categorized interpretable features according to their predominant modality, which demonstrate close correspondence to human cognitive interpretations. Our interpretability analysis in three case studies also demonstrated the great potential in understanding modality-features in gender bias, defensing adversarial attacks and controllable multimodal generation. One limitation is that we don't analyze the evaluation and results via human study. Future work may extend these methodologies to other multi-modal architectures and investigate their implications for cognitive science, ultimately fostering the development of more interpretable and human-centric AI systems.

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

# A APPENDIX

## A.1 IMPLEMENTATION FOR MONOSEMANTICITY TOOLS

The three monosemantic tools, DeCLIP, Multimodal SAE and Multimodal NCL are all on top of the canonical ViT-B-32 CLIP [7] model from OpenAI (Radford et al., 2021), with ResNet50. The four methods (including CLIP) share the same model structures but trained with different training objectives, we load them by feeding the checkpoints using the `open_clip.create_model_and_transforms` function in the published in `https://github.com/mlfoundations/open_clip`.

The feature dimensions of the output features from image encoder and text encoder are both 1024, the same for CLIP, DeCLIP and Multimodal NCL. Too retain the multimodal representation efficiency in downstream tasks, we have trained the SAE and NCL to reach very small reconstruction loss for the original features $z$ from CLIP. The dataset for NCL and SAE training is the train split (around 2900k image-text pairs) from `cc3m-wds`[8]. We train the two variants, i.e., SAE and NCL on top of the pretrained CLIP using a single 3090 GPU.

**DeCLIP.** We use the checkpoint released in `https://github.com/Sense-GVT/DeCLIP` to extract the last layer features, $z_i$ and $z_t$.

**Multimodal SAE.** We insert a SAE model to map the original feature into a sparse latent space, i.e., $z^d \rightarrow z^n$, with top-k latent as nonzero values. Empirically, we found that when $n = d$ and $k = 32$, we can get best results to balance the sparsity and downstream task performance. Such a SAE model (shared parameter) is inserted at the end of image and text encoder in CLIP. And the training loss is shown in Eq. 3.

```
def get_sae_embedding(self, z):
    z = self.encoder(z)
    z_sae = F.relu(z)
    vals, ids = z_sae.topk(self.k, dim=1)
    z_sae = torch.zeros_like(z_sae)
    z_sae.scatter_(1, ids, vals)
    return z_sae
```

Inspired by Gao et al. (2024), we train the SAE until the sparsity (the inactive dimension) of image features and text features don't increase (the same stop criteria for NCL). Noted that there are many zero values in $z^{\text{sae}}$, we remove those zero activity features (called dead latents in (Gao et al., 2024)) for the further studies. We show the changes of active dimensions of image features and text features in Figure 11.

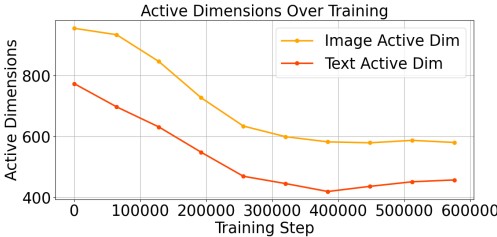

Figure 11: The changes of active dimensions over **SAE** training.

**Non-negative Contrastive Learning (NCL).** We add the NCL block, i.e., projector after obtaining $z_i$ and $z_t$ from image encoder and text encoder. The training loss is shown in Eq. 4.

---

[7] `https://github.com/openai/CLIP`

[8] `https://huggingface.co/datasets/pixparse/cc3m-wds`

```
self.projector = nn.Sequential(
    nn.Linear(embed_dim, embed_dim),
    nn.LayerNorm(embed_dim),
    nn.ReLU(),
    nn.Linear(embed_dim, embed_dim),
    )
z_ncl = self.projector(z)
```

Similarly, the activated dimensions for image features and text features decrease and are then flattened.(shown in Figure 12.) By comparing with Figure 11, we noticed that the features in SAE is much more sparse than that in NCL.

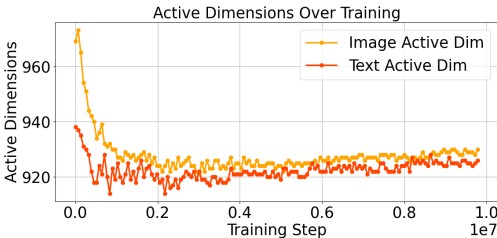

Figure 12: The changes of active dimensions over **NCL** training.

## A.2 IMPLEMENTATION OF MDS

Based on the trained CLIP, CLIP+SAE, CLIP+NCL and DeCLIP, we feed the test split of `cc3m-wds` dataset to these pretrained models, around 15k image-text pairs to calculate MDS, according to Eq.(1). The features are the last-layer output from the text and image encoder. We tried to calculate the normalization of $z_i$ and $z_t$, but found it makes little difference to the final results. It could be attributed to the existing normalization technique in image and text encoder in CLIP.

## A.3 IMPLEMENTATION AND RESULTS OF INTERPRETABILITY EVALUATION

We describe the experiment setup and other results for Section **??**.

### A.3.1 IMPLEMENTATION

**Embedding models $h$ for activated image/text samples.** Our interpretability metrics, i.e., **_EmbSim_** and **_WinRate_** are based on the embeddings of activate image/text samples by each feature. We need the embedding models to obtain these embeddings, i.e., $Z^+$ and $Z^-$. We use the Vision Transformer (ViT-B-16-224-in21k) for image embeddings and the Sentence Transformer (all-MiniLM-L6-v2) for text embeddings. The goal here is to derive the general and effective image and text embeddings, so we can also use the image encoder and text encoder from CLIP.

### A.3.2 RESULTS

**EmbSimi and WinRate for Monosemanticity measurement.** Firstly, we show the complete results for **_EmbSmi_** and **_WinRate_** in the Table 4.

**The results of monosemanticity changes as training goes on.** We show the results of monosemanticity score changes as training goes on for both NCL and SAE in Figure 13.

## A.4 IMPLEMENTATIONS AND MORE RESULTS FOR CASE STUDIES

We provide the implementation details and more experimental results for the three case studies in the follows.

Table 4: Average interpretability scores (by examining the top activated images/texts) for features extracted from VLMs.

| Models | *EmbSim* | | *WinRate* | |
|---|---|---|---|---|
| **Activated→** | Image | Text | Image | Text |
| CLIP | 0.11 | 0.45 | 0.65 | 0.59 |
| DeCLIP | 0.06 | -0.07 | 0.61 | 0.46 |
| CLIP+NCL | 0.14 | 0.45 | **0.71** | 0.60 |
| CLIP+SAE | **0.17** | **0.74** | 0.60 | **0.61** |

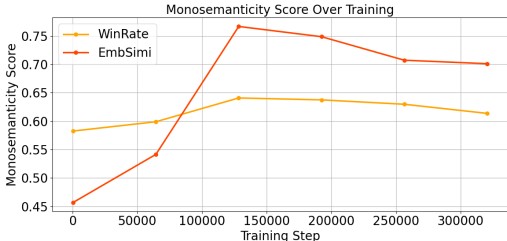

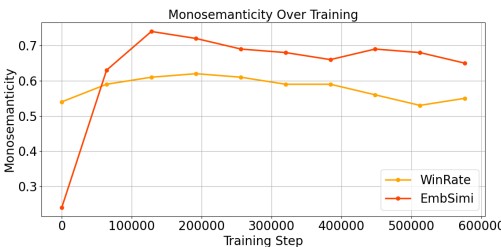

Figure 13: Monosemanticity (EmbSimi and WinRate) changes as training goes on. Upper is for CLIP+NCL, bottom is for CLIP+SAE.

### A.4.1 CASE STUDY 1: UNDERSTANDING GENDER PATTERN IN DIFFERENT MODALITIES

**Datasets.** We select male and female images using a gender classifier touchtech/fashion-images-gender-age-vit-large-patch16-224-in21k-v3 from cc3m-wds validation set. We have both input images and text, the original gender classification accuracy is 83.4% and 73.4%, respectively.

**Classification.** As the intervened features are not compatible with existing pretrained text or text classifier, we compare these features with the golden feature from male and female data. Specially, we randomly select a female/male image with classification logits larger than 0.9 (ensuring the gender patterns are obvious) as the reference features. We use the same embedding models in §A.3 , i.e., Vision Transformer and Sentence Transformer as the encoder and encode both intervened feature and golden feature. The intervened feature is labeled as the same label as the reference image with which their distance in encoder space is smaller.

**Intervention.** There are different number of `ImgD` and `TextD` for a given representation of input sample. To avoid the effects of different number of removal feature, we remove (set the corresponding dimension as zero) the minimal number between `ImgD` and `TextD`, and remove the same number of random selected features as a baseline.

**`TextD` in male concepts.** We also cluster different male descriptions according to the percentage of `TextD` features among all their top-20 activated features, and we calculate the frequency of the top7 tokens in each cluster shown in Table 6. We remove the gendered personal pronouns, e.g., he, she, woman, man, boy, girl and only focus on how gender-neutral concepts represent the gender. With more `TextD` injection, the textual descriptions become more sports related, such coach, basketball, soccer; while the sentences with less activated `TextD` have top words, as party,

| Models | CLIP | DeCLIP | CLIP+NCL | CLIP+SAE |
|---|---|---|---|---|
| *Mono* is *EmbSim* | | | | |
| **Visual Mono** | -0.007 | 0.009 | **0.043** | 0.005 |
| **Textual Mono** | -0.017 | -0.001 | **0.210** | 0.146 |
| *Mono* is *WinRate* | | | | |
| **Visual Mono** | -0.007 | 0.005 | 0.002 | **0.030** |
| **Textual Mono** | -0.069 | -0.059 | **0.018** | 0.016 |

Table 5: table
The visual and textual monosemanticity. A higher value indicates that `ImgD` captures more visual than linguistic features, and vice versa for `TextD`.

hip, game, smile, home. This trend is consistent with the social stereotype that male are more active in sport activities.

Table 6: Representative words in male-related descriptions with different percentage of `TextD`.

| Percentage of `TextD` | Top8 words in male-related textual description |
|---|---|
| 0.1 | attends, party, hip, game, comedian, city, black, artist |
| 0.12 | smile, made, blue, outside, looks, home, got, book |
| 0.18 | artist, player, film, pop, performs, festival, young, suit |
| 0.24 | player, football, basketball, team, game, portrait, holding, gym |

### A.4.2 CASE STUDY 2: DEFENSING MODALITY-SPECIFIC ADVERSARIAL ATTACKS

**Models** We employed the same ViT-B-32 CLIP as in §A.1 as the multimodality feature extractor shown in the Figure 8 to extract 1024-dimension features, so we use the categorized `TextD`, `ImgD` and `CrossD` calculated before. We use LLaVA-1.5-7b as the attacked VLM (Liu et al., 2023). The whole process of defensing adversarial attack is two steps:

- **Generating adversarial images by injecting harmful requests.** We have benign scenery image and a list of 50 harmful requests. Firstly, we create a image with white background with the text saying the one piece of harmful request, as the contrast image. Then, we apply the alignment training by minimizing the distance of benign image and contrast image in the embedding space of the image encoder. The benign image is thus being injected with harmful semantics, denoted as $\mathbf{F}_{adv}$.

- **Defensing the adversarial attacks.** To remove the toxicity of the adversarial samples, we employ the alignment training shown in Figure 8 by updating the embeddings of the adversarial samples. Specially, we only select the target features, i.e., the `ImgD`, `TextD` and `CrossD` to be involved in the training.

When attacking the VLM, we feed the adversarial images/samples along with the text prompt, i.e., the according harmful request injected to the adversarial sample. For each adversarial sample, we repeat the attack process for 100 times. For comparison, we apply the original generated 50 adversarial samples to attack VLM, and the average success rate is 73.26%; and the success rate of the (benign image - harmful request) is 10.00%. We conducted five independent runs for each experiment to ensure statistical reliability. Results in the tables show mean values across runs, with relative standard deviations below 3% for accuracy metrics.

**Computing resources cost** The experiments were conducted with a GPU with 48GB memory. Adversarial sample generation requires approximately 4 GPU hours, while adversarial sample detoxification takes approximately 6 GPU hours.

### A.4.3 CASE STUDY 3: MODALITY-AWARE CONTROL FOR TEXT-TO-IMAGE GENERATION

**Models.** We select Stable-Diffusion-v2 (https://huggingface.co/stabilityai/stable-diffusion-2) as our text-2-image generation model. As its image encoder (CLIP-

ViT-H-14-laion2B-s32B-b79K) is not the same CLIP we used before, we recalculate the MDS distribution to derive the three categorizes of features.

**More results.** We present additional images generated by modifying the original multimodal representation through feature injection from a reference image. To emphasize the distinction between `ImgD` and `TextD`, we use two reference images of horses in different backgrounds and artistic styles. Specifically, we compare two sets of images where features from sketch and oil painting styles are injected using `ImgD`. We observe that images influenced by sketches tend to be predominantly black and white, while those influenced by oil paintings appear more colorful. In contrast, the images generated using `TextD` remain visually similar across both the sketch and oil painting settings.

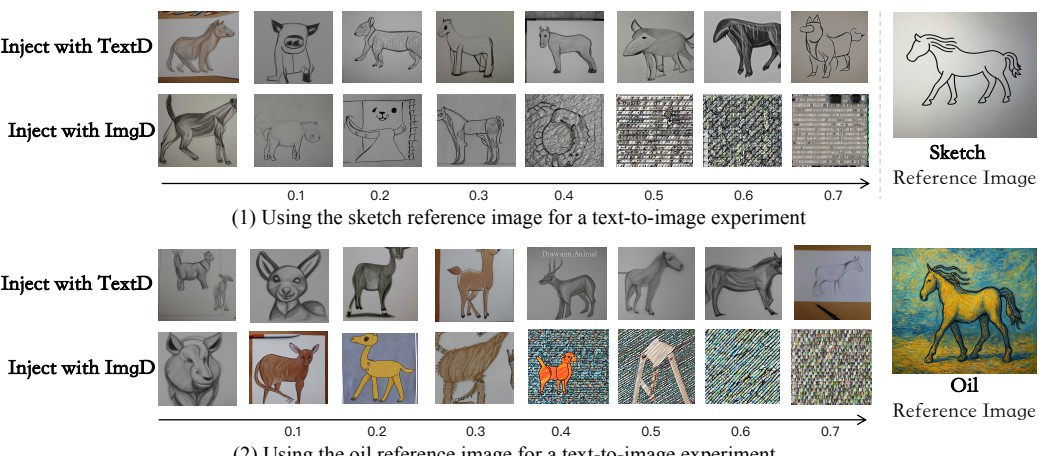

(1) Using the sketch reference image for a text-to-image experiment

(2) Using the oil reference image for a text-to-image experiment

Figure 14: Generated new images from the VLM with the text prompt *"Please draw an animal"* and varying levels of intervention from different reference image. We found that *TextD* captures significant semantic information, such as shape, etc. Notably, when sketch is selected as the reference image, both *imgD* and *TextD* display sketch-like stylistic features. When oil-painting is chosen as the reference image, both *imgD* and *TextD* exhibit styles that resemble oil paintings. Comparatively, the stylistic differences between *imgD* in conditions (1) and (2) are distinct: *imgD* in (1) lacks color, whereas *imgD* in (2) presents diverse coloration. Similar to Figure 10, *TextD* does not affect low-level visual features, while *ImgD* and show significant distortion at higher $\alpha$ values.

