# OpenReview forum: "MULTI-FACETED MULTIMODAL MONOSEMANTICITY"
_ICLR.cc/2026/Conference — Submitted to ICLR 2026_

### Official Review · Reviewer_SJnR · 2025-10-18

**Soundness:** 2
**Presentation:** 1
**Contribution:** 1
**Rating:** 2
**Confidence:** 3

**Summary:**

This paper studies how VLMs (actually only CLIP) encode modality-specific versus cross-modal information. For this, the authors introduce a Modality Dominance Score (MDS) to measure the extent to which each embedding dimension in CLIP is dominated by image or text information, or both (cross-modal). Then, they apply sparse autoencoders (SAEs) to improve monosemanticity. Then, they make case studies on gender representation, adversarial examples for VLMs and controlling text2image generation.

**Strengths:**

* important topic of study

**Weaknesses:**

* **Limited breadth:** The work is validated on a single model (CLIP), which limits the generality of the findings and makes it unclear whether the conclusions hold across architectures or datasets.
* **Limited novelty of technical contributions** The core elements (SAE, NCL) are borrowed from existing works, and the main novelty lies mostly in the metric (MDS) which is also a slight change of Liang et al. (2022) modality gap measure. Moreover, that sparse autoencoders enhance monosemanticity has already been extensively established in the literature; reproducing this finding in a multimodal setting is more of an engineering validation than a conceptual advance.
* **Poor writing and presentation quality.** The paper contains numerous typos, and the presentation feels rushed / duplicated / all over the place. I have to admit I have trouble following this study's story and red lining. It falls short of the standard expected for a top-tier conference submission. More concerns about this below.
* **missing statistical significance** or confidence intervals for key comparisons.
* **Quite anecdotal results**. I have trouble following much of the paper, maybe because of the poor presentation quality. What is missing from the results section is a clear and structured analysis of what exactly is being fused in the representation space and what remains separated. For instance, does the degree of modality fusion differ across linguistic categories, such as nouns (objects), verbs (actions), or prepositions (spatial relations), etc.? A more systematic and rigorous analysis along these lines would substantially strengthen the interpretability and explanatory power of the work. Right now, the results are very anecdotal.

### Related work:
The related work section could benefit from a rewriting. It currently looks like it was done last minute (e.g. see typos below), but my main concern is that the term "modality gap" is used *inconsistently*. In the introduction it is "a modality gap in VLM, revealing that image and text embeddings often reside in two disjoint regions of the shared embedding space.". In the related work Section 5 it is "The study of modality gaps, or the differences and limitations in how different modalities represent information, has been a topic of interest in cognitive science". This discrepancy is confusing and makes it unclear which definition the paper actually builds on. The authors should clarify and maintain a consistent interpretation throughout.

The paper discusses multimodal fusion and representation gaps, but the issue of fusion is inherently linked to that of multimodal impact: meaningful fusion presupposes that information from each modality is actually incorporated into the model’s representations. The related work section would be stronger if it acknowledged that some modality-specific information may never be taken into account at all (and is therefore absent from the representation space). Relevant prior work such as MM-SHAP (Parcalabescu & Frank, 2022) and its extensions (Wang & Wang, 2025; Goldshmidt & Horovicz, 2024; Goldshmidt, 2025) explicitly estimate each modality’s relative contribution to model outputs, offering a complementary perspective on modality integration and representational gaps.


### Typos and Grammar:

* L 011: "In this work, We" should be "In this work, we"
* L 044 missing reference to table
* L 070 "enhance modality fusion. , a persistent modality gap" please fix this
* L 122: "classficaition" typo
* L256 and L 258: Isn't it against the rules to hack the template's vspace so much? There are other vspace hacks visible in this paper, but this one is blatant and not ok.
* L 459 "polysemanticity(Olah et al., 2020)." should be "polysemanticity (Olah et al., 2020)." -- whitespace missing
* L 473 "We, instead, is to identify the different model" is not really English
* Section 6 is entitled "LIMITATIONS AND CONCLUSION". Why not conclusionS? It would make a poor paper if it had multiple limitations and one conclusion. :)
* L 479 "textual modalities. we successfully" should be "textual modalities. We successfully"
* Figure 8. benige -> benign

**Questions:**

The paper focuses on CLIP-like representations. Do you expect the same modality dominance patterns in generative multimodal models (e.g., LLaVA, or Kosmos-2, or mplug-owl-3)?

That sparse autoencoders can enhance monosemanticity is well established in prior work. It would be helpful if the authors could clarify what new insights this specific application provides beyond confirming existing findings, and how considerable engineering and computational effort is involved, and how this compares to the insight gained.

---

### Official Review · Reviewer_JXJy · 2025-10-28

**Soundness:** 2
**Presentation:** 2
**Contribution:** 2
**Rating:** 2
**Confidence:** 4

**Summary:**

The paper extracts modality specific information in VLMs and enhances the monosemanticity of VLM features with Sparse AutoEncoders and multimodal NCL loss. The modality specific features are integrated into various downstream tasks including gender bias detection, adversarial attacks and text to image generation.

**Strengths:**

The investigation of modality-specific and cross-level modality features is an important problem of multi-modal learning. The paper presents an interesting direction towards this goal and further showed its contribution towards multiple tasks. The method was overall clear and easy to read.

**Weaknesses:**

There are multiple missing details on the motivation of the method that makes the positioning of the paper unclear:
* The paper proposes modality dominance score to reflect the contribution of a modality to the d-th feature but the motivation for why this is essential. is not well discussed. Because there can be features that can be activated by both individual modalities and cross-level interactions, it is unclear why formulation in section 2.1 is justified.
* The need for multimodal NCL needs improvement as well. For example, one way to integrate multimodal to SAE is to simply use late fusion on the unimodal features and reconstruct the same.
* The performance difference is not really significant without confidence intervals to show the degradation. Further, this gap is highly specific to the task being evaluated but the downstream task is not incorporated in the proposed method.
* The inspection of features in a high dimensional space remains a challenging task, limiting the overall scalability of the proposed method.
* The clarity of parameterization of $g$ can be improved. Is $g$ shared? If yes, how is the disentanglement in individual modalities preserved? This might be a reason for the entanglement observed in sub section 3.2.2.
* The score is further averaged in eq 5 further adding to the confusion on the requirement of dissection.
* The paper further lacks an investigation of different methods and datasets (see references below), especially methods that focus on reducing the modality gap and datasets that contain these gaps. It'd be important to show if the method can capture the reduction in modality gap claimed by existing methods.

I do not consider the following points for my review but the paper writing can be improved:
* Incomplete reference to Table on Page 1
* Usage of words that are slightly vague and not self-evident. For instance within the abstract, words like data driven approach, more interpretable space  and so on. The title can be more specific as well.
* The vspaces in the paper break the reading flow. For instance, Table 1, Section 3, Section 3.2, headings of section 4 and 5 and so on.
* There are grammatical errors like focusing -> Focusing in 3.2.2, such colorful shifts in section 4.1.

References.
[1] Goyal et al. Making the V in VQA Matter: Elevating the Role of Image Understanding in Visual Question Answering.
[2] Dancette et al. Beyond Question-Based Biases: Assessing Multimodal Shortcut Learning in Visual Question Answering.
[3] Si et al. Language Prior Is Not the Only Shortcut: A Benchmark for Shortcut Learning in VQA.
[4] Tong et al. Cambrian-1: A Fully Open, Vision-Centric Exploration of Multimodal LLMs.
[5] Madaan et al. Jointly Modeling Inter- & Intra-Modality Dependencies for Multi-modal Learning.
[6] Wang et al. An Information Criterion for Controlled Disentanglement of Multimodal Data.
[7] Schrodi et al. Two Effects, One Trigger: On the Modality Gap, Object Bias, and Information Imbalance in Contrastive Vision-Language Models.
[8] Makino et al. Detecting incidental correlation in multimodal learning via latent variable modeling.

**Questions:**

Please refer to my comments above.

---

### Official Review · Reviewer_i7vU · 2025-10-31

**Soundness:** 3
**Presentation:** 3
**Contribution:** 2
**Rating:** 4
**Confidence:** 3

**Summary:**

The paper introduces the Modality Dominance Score (MDS) to analyze modality-specific dimensions in vision-language models It identifies image-, text-, and cross-dominant features and uses monosemanticity methods (SAE, NCL) to interpret them. Applications include bias analysis, adversarial robustness, and modality control.

**Strengths:**

* Relevant and important topic on VLM interpretability.

* Clear formulation of MDS and integration with monosemanticity tools.

* Interesting applications demonstrating practical value.

**Weaknesses:**

* Experiments are conducted on CLIP models, raising questions about how results transfer to current larger, instruction-tuned VLMs (e.g., LLaVA, QwenVL)

* The work share similar findings with previous works without proper mentions or citations. The work.should position itself poperly to previous works that explored similar questions. In particular, the modality gaps, multimodal features and concepts [1, 2, 3]

* I find most of findings lack novelty. For instance, using SAE to enhance monosemanticity, and extracting modality-specific and multimodal neurons.

[1] Schwettmann, Sarah, et al. "Multimodal neurons in pretrained text-only transformers." Proceedings of the IEEE/CVF International Conference on Computer Vision. 2023.
[2] Parekh, Jayneel, et al. "A concept-based explainability framework for large multimodal models." Advances in Neural Information Processing Systems 37 (2024): 135783-135818.
[3] Shukor, Mustafa, and Matthieu Cord. "Implicit multimodal alignment: On the generalization of frozen llms to multimodal inputs." Advances in Neural Information Processing Systems 37 (2024): 130848-130886.

**Questions:**

Please check the weakneses section.

---

### Official Review · Reviewer_ZiJQ · 2025-11-01

**Soundness:** 3
**Presentation:** 3
**Contribution:** 3
**Rating:** 4
**Confidence:** 2

**Summary:**

This paper investigates the internal representations of multimodal models and introduces a novel framework for analyzing and enhancing their interpretability. Specifically, the authors propose the Modality Dominance Score (MDS), a metric that quantifies how strongly each feature dimension in a model is dominated by either the visual or textual modality. Based on this measure, features are categorized as text-dominant, image-dominant, or cross-modal. To further enhance monosemanticity and disentanglement, the paper combines Sparse Autoencoders (SAE) and Non-negative Contrastive Learning (NCL), enabling more interpretable and modality-specific representations. The authors also design new quantitative metrics (EmbSim and WinRate) to evaluate interpretability without relying on external judges. Extensive experiments on CLIP-based architectures demonstrate improved modality separation, semantic clarity, and robustness. Through case studies in bias detection, adversarial defense, and controllable text-to-image generation, the paper highlights both the interpretability and practical value of its approach. Overall, it provides a compelling framework for understanding and leveraging modality-specific behaviors in large vision-language models.

**Strengths:**

1. The paper introduces a novel quantitative framework for analyzing multimodal representations via the Modality Dominance Score (MDS), which systematically disentangles features into image-dominant, text-dominant, and cross-modal groups.
2. he case studies (bias detection, adversarial defense, controllable text-to-image generation) convincingly demonstrate that the disentangled features are not only interpretable but also actionable in downstream applications.

**Weaknesses:**

1. This paper is not written well. Some sentences are not clear and there are many typos. For example, in abstract line 11, "We take a data-driven approach to address this question", what question do you want to address? Such unclear description happens several times in main paper, which makes me feel hard to follow.
2. Section 2.3 is not make sense. To evaluate modality-specific information, only selecting 2 samples is obvious not promising to me. You can always "select" some cases that provide some meanings in your setting, but without statistics and quantitative analysis, it is not very convincing. For greater promising, I recommend complementing this qualitative analysis with statistical measures, for example, computing intra-group embedding variance, feature–modality mutual information, or cross-modal activation correlation, to verify that ImgD, TextD, and CrossD features differ significantly in their semantic consistency and modality dependence.
3. The monosemanticity evaluation relies on selecting the top-20 most-activated samples per feature, but the choice of 𝑚=20 is fixed and unexamined. Since the interpretability scores (EmbSim and WinRate) may be sensitive to this threshold, it would strengthen the validity of the analysis to include an ablation study varying m (e.g., 10, 30, 50). This would clarify whether the observed monosemanticity improvements of SAE and NCL are robust to the sample selection parameter.
4. The paper does not include quantitative comparisons with existing interpretability or modality-disentanglement methods. All evaluations (e.g., EmbSim, WinRate, and MDS-based analyses) are performed within the authors’ own framework, mainly comparing CLIP, DeCLIP, and their proposed post-hoc variants (SAE/NCL). While this effectively isolates the effect of their techniques, it leaves unclear how the proposed metrics relate to prior interpretability benchmarks or human evaluations. Including quantitative cross-method comparisons would considerably strengthen the empirical validity of the claims. It would be great if the authors can explain why the don't (need) compare with previous work, otherwise, providing some quantitative comparison is more convincing for this work.

Overall, I am open to be challenged by other reviewers or AC or authors. I am open to change my score if authors can solve all my concerns.

**Questions:**

1. What is the table in line 044?
2. In Section 2.2, when features from different modality groups are removed, the performance drop caused by deleting CrossD features is the smallest. However, since CrossD features represent the shared common information between image and text, one might expect that removing them would cause a larger performance degradation — for example, in image classification, removing shared cross-modal features should disrupt the semantic grounding between image and text more than removing purely textual ones. Why, then, does the paper show that deleting CrossD features leads to less performance drop than deleting TextD or ImgD features?
3. In Section 3.1, the authors include DeCLIP—a separately trained model with a different training objective—as part of their comparison for feature disentanglement and monosemanticity. Could the authors clarify the rationale behind using DeCLIP here? Since DeCLIP alters the pretraining procedure rather than applying a post-hoc disentangling method to CLIP features (as done with SAE and NCL), this comparison may conflate training-level improvements with representation-level interpretability. How do the authors ensure the fairness and validity of this baseline within the context of Section 3’s analysis?
4. How many samples are you selected for experiments CASE STUDY 1? It is not very intuitive that "feminine concepts are primarily preserved in ImgD, whereas male concepts are more affected by the removal of TextD", although according to the table it is. Could author provide a more detailed analysis on this conclusion? It is not intuitive for me because female/male are parallel concepts and they are homogeneous, so they should at least follow the same interpretable pattern. I'd like to see a clearer explanation about this conclusion.

---

### Meta-Review · Area_Chair_e2UX · 2025-12-22

**Summary:**

This paper explores the representations of large-scale multimodal models by analyzing interpretable, monosemantic features. In their review, all reviewers provided negative ratings. Their concerns included limited novelty and impact (e.g., the technical components are already established, and the authors applied them in an engineering capacity rather than introducing new insights), poor presentation (e.g., reviewers highlighted the difficulty of following the narrative), weak methodology, and generalization issues (e.g., the experiments focus solely on CLIP).

**Reviewer Concerns:**

The authors did not provide rebuttal.

**Reviewer Scores:**

The authors did not provide a rebuttal, so the reviewers were not involved in the discussion.

---

### Decision · Program_Chairs · 2026-01-26

Reject